# Test and Analysis of Vegetation Coverage in Open-Pit Phosphate Mining Area around Dianchi Lake Using UAV–VDVI

**DOI:** 10.3390/s22176388

**Published:** 2022-08-24

**Authors:** Weidong Luo, Shu Gan, Xiping Yuan, Sha Gao, Rui Bi, Lin Hu

**Affiliations:** 1School of Land and Resources Engineering, Kunming University of Science and Technology, Kunming 650093, China; 2Plication Engineering Research Center of Spatial Information Surveying and Mapping Technology in Plateauand Mountainous Areas Set by Universities in Yunnan Province, Kunming 650093, China; 3College of Geosciences and Engineering, West Yunnan University of Applied Sciences, Dali 671000, China

**Keywords:** UAV, VDVI, RF, FVC

## Abstract

This work aimed to detect the vegetation coverage and evaluate the benefits of afforestation and ecological protection. Unmanned aerial vehicle (UAV) aerial survey was adopted to obtain the images of tailings area at Ma’anshan near the Dianchi Lake estuary, so as to construct a high-resolution Digital Orthophoto Map (DOM) and high-density Dense Image Matching (DIM) point cloud. Firstly, the optimal scale was selected for segmentation by considering the terrain. Secondly, the visible-band difference vegetation index (VDVI) of the classified vegetation information of the tail mining area was determined from the index gray histogram, ground class error analysis, and the qualitative and quantitative analysis of the bimodal index. Then, the vegetation information was extracted by combining the random forest (RF) classification algorithm. Finally, the extracted two-dimensional (2D) vegetation information was mapped to the three-dimensional (3D) point cloud, and the redundant data was eliminated. Fractional vegetation cover (FVC) was counted in the way of surface to point and human–machine combination. The experimental results showed that the vegetation information extracted from the 2D image was mapped to the 3D point cloud in the form of surface to point, and the redundant bare ground information was eliminated. The statistical FVC was 36.06%. The field survey suggested that the vegetation information in the turf dam area adjacent to the open phosphate deposit accumulation area research area was sparse. Relevant measures should be taken in the subsequent mining to avoid ecological damage caused by expanded phosphate mining. In general, applying UAV measurement technology and related 2D and 3D products to detect the vegetation coverage in an open phosphate mine area was of practical significance and unique technical advantages.

## 1. Introduction

Yunnan province is rich in phosphate mineral resources. More than one third of the discovered phosphate ores are accompanied by natural radionuclides such as uranium, thorium, and radium. The rapid development of many industries, such as agriculture, forestry, and industry, increases the amount of phosphate mining, and aggravates the harm of radioactive elements to the surrounding vegetation. Fractional vegetation cover (FVC) is a parameter characterizing vegetation and an important parameter in ecology, geography, and remote sensing. Accurate and timely acquisition of vegetation cover information is the common demand of all scientific fields, which is of important scientific significance. Based on the field measured data, including the remote sensing data and precipitation data, Chen Jianjun [1] et al. analyzed the spatial distribution characteristics of vegetation coverage change in the cold grassland on the Qinghai–Tibet Plateau, and revealed that the area with significant change and non-significant change in the alpine grassland were 17.86% and 80.72%, respectively. To monitor and evaluate the habitat quality of Asian elephants in Sri Lanka, Wu Linlin [2] et al. discussed the numerical distribution of habitat quality combining altitude and vegetation coverage gradient, and concluded that the habitat quality was higher, vegetation coverage was higher, and protected areas were more densely distributed in the south.

Compared with other methods, unmanned aerial vehicle (UAV) remote sensing shows short period and high resolution, so it can adapt to the rapid acquisition and updating of vegetation coverage in small areas. He Haiqing [3] et al. extracted the soybean coverage by the rest of the features of interference by relying on the two-dimensional (2D) and remote sensing image technologies combined with the three-dimensional (3D) point cloud coverage of the soybean extraction method; the results showed that this method is suitable for the complex background of flower bud differentiation period coverage to extract soybean, gamma enhancing the index of green leaf vegetation extraction accuracy can be improved, and the overall accuracy of coverage extraction combined with 3D point cloud information is more than 98%. Yang Xu [4] et al. comprehensively considered the advantages of 3D image point cloud, Hue, Saturation, Intensity (HIS) and matching point cloud, and applied them to land cover type recognition and extraction. Li Pengfei [5] et al. selected the visible-band difference vegetation index (VDVI) suitable for the extraction of slope vegetation of the dump in the research area through the sample survey method and UAV remote sensing and visible vegetation index (VI) calculation, and estimated its vegetation coverage. In order to study the vegetation coverage on visible images, Xie Bing [6] et al. constructed a Digital Orthophoto Map (DOM) by UAV aerial images, and found that the overall accuracy of vegetation coverage extraction reached 93.5% by adopting the red–green–blue (RGB) ratio VI (RVI). Wang Meiqi et al. [7] took xiyuan No.4 Coal Mine administration area as the research area for the flight experiment, collected image data by UAV, converted RGB color space of image into Hue–Saturation–Value (HSV) color space, and limited the value range of H and S values, aiming to analyze the vegetation coverage of abandoned mines; the results showed that UAV remote sensing technology can quickly extract vegetation and calculate vegetation coverage, with an error of lower than 6.86%.

In summary, statistics of vegetation coverage based on 2D images has been focused on for many times. Based on the cm-level high-precision DOM constructed from UAV aerial images, 2D images were segmented in accordance with the optimal segmentation scale, and then the exponential gray histogram, ground class error analysis, and bimodal index were analyzed qualitatively and quantitatively. In addition, the visible light vegetation index (VDVI) of the classified vegetation information of the tailing area was determined, and the vegetation information was extracted by combining the selected ground class sample information and the RF classification algorithm. Finally, the extracted 2D vegetation information was mapped to the 3D point cloud by direct linear transformation, and the redundant bare soil information covered by the mapping data was eliminated. Based on the VDVI, the vegetation coverage of plateau open phosphate mine area can be accurately detected in the point cloud in the surface-to-point mode.

## 2. Research Area and Data Preparation

### 2.1. Research Area

The research area referred to the Ma’anshan near the outlet of Dianchi Lake in Yunnan Province, as shown in Figure 1. The whole research area was in the plateau mountain area, with a total area of 0.40 km^2^. The terrain in the research area is steep and belongs to the typical plateau mountain landform. The tailings pond is made of artificially continuously adding solid tailings slag on the basis of the low dam. If the dam body is sufficiently stable, tailings or other industrial wastes with low phosphorus content obtained after the selection of phosphate rock can be stored in the crest area. The mixed tailings and industrial wastewater are discharged to the tailings flood discharge area in liquid form, and then the water is dried in the sun, and finally accumulated in the tailings sand storage area in solid form [8].

The vegetation conditions for subsequent research area was one of the important indicators of ecological condition; the research area was lush green vegetation surrounding the adhesion of the cordillera, but the mining area for internal mining research was more bare. There was a small hill because it had not been mined, or mining had not continued for a long time, and there was a little low vegetation attached. To sum up, the external climate in the research area was suitable for surface characteristics investigation using UAV.

### 2.2. Data Acquisition

The experimental data were collected on 3 October 2020. Image data in the research area was acquired by DJI Phantom 4 RTK using dynamic post-processing differential technology. During the data acquisition, route planning for the research area was carried out according to the actual geographical environment and traffic conditions by combining with the Google Earth platform. Then, appropriate parameter settings were selected based on the route planning in the early stage and data quality requirements in the later stage. As the data acquisition was mainly oriented to the analysis and application of vegetation information in the open-air phosphate mining area around Dianchi Lake, the flight parameters were set as follows: heading overlap degree was 80%, side overlap degree was 70%, and average flight altitude was 250 m. The weather conditions of this aerial survey were good, and two sorties were flown. In total, 155 images were collected, with an average resolution of 0.07 m. The specific route planning in shown in Figure 2.

### 2.3. Construction of the DIM Point Cloud and DOM

#### 2.3.1. Construction of the DIM Point Cloud

The DIM point cloud and LiDAR point cloud were compared, as shown in Table 1. Characteristics of the DIM point cloud are summarized as follows. ① Compared with a 3D laser scanner, the UAV image data acquisition equipment showed lower cost, easier operation, and higher efficiency. ② The DIM point cloud data contained some 3D coordinate information, rich spectral information, and texture information, providing full information for subsequent processing. ③ The DIM point cloud could not realize echo strength, so it could not collect the surface data directly from vegetation, buildings, and other ground objects compared with LiDAR point cloud. ④ Due to the existence of ground objects, the DIM point cloud can directly obtain the digital surface model.

The DIM point cloud contained spectral information of RGB in three visible bands. Compared with the LiDAR equipment, UAV showed the advantages of low price and easy portability, based on which the DIM point cloud was widely applied in various industries. Semi-global matching (SGM) algorithm [9] obtained the DIM point cloud, and its energy function is shown in Equation (1). At the minimum, the basic process of finding the optimal parallax of each pixel by global energy optimization strategy mainly consisted of four steps: cost matching calculation, cost aggregation, parallax calculation, and parallax refinement.
(1)$$E\left(D\right)=E_{data}\left(D\right)+E_{smooth}\left(D\right)$$

In the above equation: $E_{data}$ referred to the data item to represent the overall matching cost measure; $E_{smooth}$ was the smoothing term, so that the parallax could meet the condition constraint.

Based on PhotoScan, 384,401,447 DIM point clouds were constructed, and the density of point cloud was 214.36 points/m^2^, as shown in Figure 3. As revealed in Figure 3a,b, the whole research area was completely constructed without point cloud holes, and the texture of lawn dam was clear. The ground features (as shown in Figure 3c,d) disclosed that the tailings areas showed obvious contours, and the image tone in area 1 was darker than that in area 2, indicating that the water content in in area 1 was more. The constructed DIM point cloud can clearly judge the ground object boundary between tailings and bare land and vegetation only by visual observation.

The DIM point cloud was employed to construct a digital surface model, and the section of the research area was drawn from the northwest, west, and southwest directions based on 3D in ArcGIS. As shown in Figure 4, the elevation difference between the tailing sand accumulation area and the bottom of the lawn dam was 140 m, and the lawn dam presented a stepped distribution. The topographic characteristics of the field should be considered in subsequent segmentation tests.

#### 2.3.2. Construction of the DOM

The DOM covered attribute features and geometric features with full content, which can directly express the research area truly. DOM is constructed based on the automatic point conversion of POS data, fine solution of external position elements, and digital differential correction. The specific results are shown in Figure 5a,b. As two sorties were flown, the DOM of sorties 1 and 2 should be constructed for image matching and image fusion, as shown in Figure 5c. Finally, fusion chromatic difference was eliminated according to color space uniformity, and the DOM with 21,715 × 18,108 pixels and resolution 0.08 m was completely constructed, as shown in Figure 5d.

#### 2.3.3. 2D Image Information and 3D Point Cloud Mapping

A machine learning algorithm was adopted to classify and extract the target ground objects in UAV images. The extracted classification results were 2D pixel coordinates, and the mapping relationship between the images and the DIM point cloud was established. The results of the 2D image classification and extraction can be directly mapped to the 3D point cloud to obtain the target data of the DIM point cloud. The methods for establishing the mapping relations have been summarized in many domestic and foreign studies [10,11]. ① The direct solution method aimed to realize the linear transformation through the image point coordinates of feature points in the 3D image and the feature point coordinates of the 3D point cloud according to the collinear equation. ② In the point cloud transformation method, the 3D point clouds were obtained through intensive matching and 3D reconstruction, and the mapping between the point clouds was established based on iterative closest point (ICP) and other algorithms. ③ The image conversion method required us to convert the point cloud from the 3D to the 2D image; the mapping relationship with the camera image could be established through 2D image matching.

To guarantee the mapping accuracy, an evaluation model can be construction through (1) removing the remaining ground class data and retaining 2D vegetation vector data, counting the number of planes of vegetation vector data (represented by M), counting the number of mapped vegetation information surfaces in the 3D point cloud (denoted by N), and calculating the mapping error of vegetation information (*Q_i_*) with Equation (2):(2)$$Q_{i}=\frac{N}{M}\times 100\%$$

## 3. Research Methods

### 3.1. Segmentation and Classification

#### 3.1.1. Optimal Segmentation Scale

Image segmentation was critical before image analysis, which was widely used in geography, medicine, biology, and many other fields. Image can be segmented by three methods: region-based segmentation method, edge-based segmentation method, and threshold-based segmentation method. There was no general method suitable for image segmentation in the proposed method, so it had to be combined with research experience and actual situation to meet the specific segmentation. The image resolution was gradually improving, and the differences in various ground objects in the image were becoming more and more obvious, so it was difficult to process all ground objects with simple segmentation. Multi-scale segmentation was a bottom-up network structure that calculated the homogeneity value of adjacent pixels based on a single pixel and merged them into larger objects if the homogeneity value was lower than the specified threshold. This cycle was carried out until the pixel-point combination stopped at the specified scale [12], as shown in Figure 6.

Multi-scale segmentation is region-based. The specific segmentation parameters calculated as shown in Figure 7 should meet the following conditions. Firstly, appropriate segmentation scale should be selected according to the actual size of the ground objects. Secondly, when boundary of the ground objects showed smooth area, the weight of smoothness was significant, while the weight of compactness was small. On the contrary, when boundary of the ground objects showed prominent edges or corners, the weight of smoothness was small, while the weight of compactness was significant. Then, the color of the spectral factor should be greater than that of the shape factor, avoiding the under-segmentation, but the over-segmentation may partially appear.

The segmentation parameters were set. After the segmentation scale was determined, the weights of color and shape were determined according to the consistency criterion, in which color was mainly RGB band parameters, and shape mainly referred to smoothness and compactness.

The segmentation consistency standard C  was mainly defined by $h_{color}$ and $h_{shape}$, which can be calculated with Equation (3) [13]. $h_{color}$ referred to the setting of spectral factor, which was mainly calculated according to Equation (4), while $h_{shape}$ was defined according to Equation (5).
(3)$$C=w_{color}\times h_{color}+w_{shape}\times h_{shape}$$
(4)$$h_{color}=\sum_{c}w_{c}+\sigma_{c}$$
(5)$$h_{shape}=w_{smoothness}\times h_{smoothness}+w_{compatness}\times h_{compatness}$$
where $w_{color}$ and $w_{shape}$ represented the weight of spectrum and shape, respectively, and their sum was equal to 1. The number of image layers was represented by c; $w_{c}$ represented the weight of image layer; $\sigma_{c}$ was the standard deviation of image layer; $w_{smoothness}$ and $w_{compatness}$ were weights on smoothness and compactness, respectively, and their sum as equal to 1.

$h_{smoothness}$ and $h_{comothness}$ were set to optimize the results of image objects and make their shape more reasonable. The specific calculation equations are shown as follows.
(6)$$h_{smoothness}=n_{merge}\times \frac{l_{merge}}{b_{merge}}-\left(n_{obj1}\times \frac{l_{obj1}}{b_{obj1}}+n_{obj2}\times \frac{l_{obj2}}{b_{obj2}}\right)$$
(7)$$h_{compatness}=n_{merge}\times \frac{l_{merge}}{\sqrt{n_{merge}}}-\left(n_{obj1}\times \frac{l_{obj1}}{\sqrt{n_{obj1}}}+n_{obj2}\times \frac{l_{obj2}}{\sqrt{n_{obj2}}}\right)$$
where l referred to the actual edge length of the object, b was the shortest side of the object, and n indicated the area of the object. The weight of $h_{smoothness}$ was high, and the boundary of the segmented object was smoother. If the weight of $h_{smoothness}$ was high, the split object shape was similar to the rectangle.

In multi-scale segmentation, the optimal scale was of crucial significance to extracting and analyzing the image information. At present, discriminant index method, subjective experience method, and calculation model method are available for the selection of optimal scale. There is a lot of human intervention in discriminant index and subjective experience. There are four methods for computer model, including local variance method, maximum area method, distance index method, and mean variance method.

#### 3.1.2. Random Forest (RF)

RF algorithm is important in image classification, which features high robustness, good compatibility, and good anti-noise ability. It can effectively avoid excessive image fitting and assist the rest of the raster data. In addition, the image should be processed after segmentation. The essence of RF is to combine a series of individual decision trees $\left\{h\left(X,\theta_{n}\right),n=1,2\ldots m\right\}\in \left\{true,false\right\}$ to become a classifier, and $\left\{\theta_{n}\right\}$ is an independent and uniformly distributed random vector.

As an independent and uniformly distributed random vector, decision tree can select the optimal feature through recursion, and consists of three nodes: root, middle, and leaf. The blue circle in Figure 8 represented the root node, and the red rectangle represented the leaf node, relying on which the classifier is formed in RF algorithm. The following steps had to be taken to establish the decision number. ① The part of the data to be trained were placed on the root node, and then the training set was divided into many subsets through optimal feature selection. ② The correctly classified subsets were mapped to leaf nodes one by one to complete the construction of leaf nodes. ③ Step 1 was repeated for the improperly classified subset. Step 2 was required for all the other training sets except for a few sets with no suitable feature selection to classify the subset correctly. ④ Starting from the root node, only one leaf node, i.e., subset, has a definite class, to complete the construction of decision tree.

Decision tree is a single classifier model, which can only obtain local optimal solutions rather than global ones. Ensemble learning methods such as Bagging, Random Subspace, and Boosting were added to train multiple classifiers in this work.

The process of RF algorithm mainly included the generation of training sets, the construction of decision trees, and the execution of the algorithm. As shown in Figure 9, for a forest with a scale of N, N decision trees were required to produce the same number of training sets. In addition, Bagging was incorporated to randomly sample the N training samples, which may lead to duplicate sampling of some samples. CART was undertaken as a classifier in RF, where Gini index of each attribute was calculated in the training of CART [14]. Nodes were classified by selecting the minimum Gini index, and a decision tree was established according to the recursive form. The calculation process of splitting nodes based on the minimum Gini impurity criterion was shown in Equation (8).
(8)$$Gini\left(t\right)=1-\sum_{j=1}^{c}{\left[p\left(j|t\right)\right]}^{2}\;\;j=1,\ldots \ldots ,c$$
where p(j|t) referred to the probability of category *j* on the node *t*. If all the samples of node *t* belonged to the same category, Gini() was the regional minimum, suggesting that the samples were the purest, while the Gini() index was the maximum value 1 if all samples on node *t* were not of the same category.

If the sample was divided into M branches, the Gini index of node splitting was calculated according to Equation (9).
(9)$$Gini\left(x\right)=\sum_{i=1}^{m}\frac{n_{i}}{n}Gini\left(i\right)$$
where *m* was the number of nodes, $n_{i}$ referred to the number of samples of node i, and n indicated the number of upper-layer node samples.

RF algorithm was an ensemble classifier model formed by comprehensively considering multiple decision trees. It was not only used for classification but also for solving regression. Its voting decision-making process was shown in Equation (10).
(10)$$H\left(x\right)=arg\;\underset{Y}{\max}\sum_{i=1}^{K}I\left(h_{i}\left(x\right)=Y\right)$$
where H(x) was the combined classification model, I was an indicative function, and the type with the most votes was judged as the output result based on the maximum voting rule; $h_{i}$ was a single decision tree, and Y was the output variable.

### 3.2. Vegetation Index of Visible Band

Normalized difference vegetation index (NDVI) was to measure the ground profile efficiently in the remote sensing technology. Because vegetation showed different reflectance and absorption rate of various spectra, different spectral bands can be combined to detect the vegetation characteristics. With the increasing progress of science and technology, there are more band combination methods to study the vegetation index, but most of them needed the participation of near-infrared bands, such as NANI and RVI. Only combining with visible bands can identify relatively few VIex. Yahui Guo [15] et al. used UAV and SPAD-502 for long-term observation of corn yield, and they derived the modified red and blue VI (MRBVI) of visible bands based on the observation results, so as to monitor corn growth and predict corn yield. Saponaro Mirko [16] et al. studied the influence of the spatial resolution of the DOM produced by UAV after studying the VI of visible band in different UAV technologies. The results showed that the triangular greenness index (TGI) VI of the visible band combination was highly separated from the vegetation and non-vegetation regions. Gao Sha [17] et al. put forward the view of mountain vegetation types of the most widely distributed landscape matrix by building the VDVI and based on integration of spectral information point cloud data of feature extraction of typical vegetation research, and they found that the number of UAV image data using VDVI extract vegetation showed high accuracy, without obvious errors or omissions. Combined with the research of domestic and foreign scholars, main combinations of VIex in visible bands were listed as shown in Table 2.

### 3.3. Precision Evaluation

The consistency between classified results and the actual ground objects was called the classification accuracy, which evaluated the degree to which the classified results meet the actual situation. The classification accuracy can be evaluated qualitatively and quantitatively. The qualitative evaluation was based on visual interpretation of workers, which was interfered with by human factors. The quantitative evaluation is based on the reference index. In this section, the classification results of DOM constructed from images in the research area were evaluated by combining manual intuitive judgment of the two classification methods with reference indexes, and the accuracy was expressed by the percentage.

Quantitative evaluation can be based on the confusion matrix of N rows × N columns, where N referred to the total number of image classification, row as all the classification results. In addition, the matrix was used for classification error analysis. The erroneous and missed judgments of the confusion matrix were to calculate the classification accuracy. Erroneous judgment referred to the error caused by a category being wrongly judged, while missed judgment referred to the error caused by the omission of a category. In addition, the confusion matrix can obtain the classification accuracy. As shown in Equation (11), probability of being consistent with the classification result was the value obtained by comparing the main diagonal element correctly classified in the matrix with the same category of column through training the sample data of the ground class. User accuracy, as shown in Equation (12), was compared with the actual category of the samples randomly selected from the classification results, and the probability of coincidence is the value obtained by comparing the main diagonal elements correctly classified by the matrix with similar rows. Overall accuracy (*OA*), as shown in Equation (13), was the ratio of the number of correctly classified samples to the samples used, which was mainly the sum of diagonal elements obtained by comparing with all elements. *OA* value was not the accuracy of a certain category. If the *OA* value was large, the accuracy of a category may be low; otherwise, the accuracy of a category may be relatively high when the *OA* value was small. The *Kappa* coefficient, as shown in Equation (14), referred to the degree of comparison and consistency between the classification image map and the reference classification map, and its value was [0, 1]. The higher was the Kappa coefficient value, the higher was the classification accuracy.
(11)$$PA=\frac{a_{ii}}{a_{ti}}\times 100\%$$
(12)$$UA=\frac{a_{ii}}{a_{it}}\times 100\%$$
(13)$$OA=\frac{\sum_{i=1}^{n}x_{ii}}{N}\times 100\%$$
(14)$$kappa=\frac{N\sum_{i=1}^{n}a_{ii}-\sum_{i=1}^{n}\left(a_{ti}+a_{it}\right)}{N^{2}-\sum_{i=1}^{n}\left(a_{ti}+a_{it}\right)}$$

In the equation above: $a_{ii}$ was the number of correctly classified items in class I; $a_{ti}$ was the total number of classes I in all samples; $a_{it}$ referred to the total number of categories I in the result after completion of classification; $x_{ii}$ represented the number accurately classified in all samples; and *N* was the total number of sample sets [24].

## 4. Test and Result Analysis

### 4.1. Extraction of Vegetation Information from 2D Images

The DOM segmentation image with 21,715 × 18,108 pixels and a resolution of 0.08 m mainly faced the vegetation information of the open-pit phosphate mining area around Dianchi Lake, so the local variance method was employed to select the optimal segmentation scale. The rate of change in LV (LV − ROC) curve can be obtained by using the local variance method for image segmentation. The curve corresponding to the segmentation scale of local variance presented multiple peaks, which correspond to the segmentation scales of different ground objects.

The segmentation in this work was mainly about the low-rise vegetation information of the plateau tailings area, based on eCognition and DOM data. Firstly, the initial segmentation scale of the first and third layers was 10, 5, and 1, respectively, and the step size was 10. Secondly, the local variances of the segmentation object were counted to obtain the LV − ROC curve, as shown in Figure 10a. The curvature peak corresponded to multiple optional segmentation scales. Finally, elevation and slope were added into the segmentation to determine the optimal scale. When the elevation was added, it can be judged that the optimal scale corresponding to the peak was concentrated below 500, as shown in Figure 10b. As a result, when the slope was added, as shown in Figure 10c, there was only one prominent peak, so the optimal scale corresponding to the peak was 440 in this work.

Based on the optimal segmentation scale of 440, shape was determined for selection of smoothness and compactness. Since the boundary of the research area was not smooth, and the sum of color and shape was 1, this work aimed to classify the vegetation information of the open-pit phosphate ore area. In total, 2935 objects were segmented by selecting compactness as 0.6 and smoothness as 0.4. The results are shown in Figure 11.

The tailing sand and a small part of bare soil mixed area, R_1_, completely divided the tailing sand area near the bare soil, and R_2_ and R_3_ were the mixed distribution areas of vegetation and bare soil. In R_2_ and R_3_, the segmentation effect of vegetation and bare soil was smooth, and the segmentation boundary was smooth. Based on the visual interpretation, it can be judged that no other land groups were mixed in a single land group.

ArcGIS Pro was employed to calculate the 9 visible VIex through visible band according to the equations in Table 2, and the gray histogram of each index was obtained, as shown in Figure 12. In NGRDI, NGBDI, MGRVI, RGBVI, ExG, VDVI, and EGRBDI, the bright color region represented the vegetation information, and the relatively dark color region showed the non-vegetation information such as bare soil and tailing sand. The above VDVI could show the vegetation information well, but the gray value of NGBDI vegetation was close to that of bare soil, so that they were easy to be confused in the subsequent classification. The dark areas in MRBVI and CIVE represented the vegetation information, and light areas represented the non-vegetation information. CIVE was more adequate than MRBVI in identifying the vegetation area in turf dam area. Brightness of vegetation in the left part of tailing sand and that in the lawn dam was different, and it was easy to cause mistakes and omissions in classifying the vegetation information. Therefore, it was necessary to classify them separately and then categorize them as vegetation.

To better analyze the extraction results of nine visible VIex, the mean error and standard deviation of each VI in the research area with four types of typical ground objects were counted, and the specific values are shown in Table 3. The theoretical range values of CIVE and EGRBDI were [18, 19] and [0, 1], respectively, and those of the other seven visible VIex were [−1, 1]. The theoretical intervals were different, so the standard deviations of the three types of ground objects were mainly analyzed. The standard deviations of MGRVI, MRBVI, RGBVI, and ExG were greater than 0.1, and that of MGRVI was 0.148, which was the maximum. The standard deviations of CIVE, EGRBDI, and VDVI were the smallest, while those of CIVE, EGRBDI, and VDVI were smaller in bare land and tailing sand. Therefore, priority was given to the above three visible VIex when UAV images were adopted to identify the vegetation information of the open-pit phosphate ore area around Dianchi Lake.

To determine the most suitable visible VI for this test, the number of pixels in the gray histogram was counted. If there was an index with obvious double peaks, the corresponding difference and performance of different ground classes were relatively better, as shown in Figure 13. The abscissa and ordinate represented the value of VI and the number of pixels, respectively. The CIVE, EGRBDI, and VDVI were analyzed based on the statistical analysis of the error of VI. Double peaks were found in Figure 13g–i. Although the distance between double peaks of VDVI was very close to each other, some pixels of CIVE and EGRBDI exceeded the theoretical range, while NGRDI, MRBVI, and MGRVI were excluded because no double peaks were found. Combined with number gray histogram analysis, statistical classification error analysis, and bimodal index, the VDVI which was most suitable for the classification vegetation information of the tailing mining area was selected.

For extraction of vegetation information, it should input images with the optimal segmentation scale 440 firstly based on the eCognition, and then extract the appropriate or required information in the sample selection tool. This work aimed for statistical FVC, grass and shrubs were included in the vegetation. The selected class sample information was sourced from the whole research area, as shown in Figure 14.

Secondly, this work was mainly to classify the low vegetation information in the tailing area of the plateau, so the RF algorithm was mainly employed to adjust the four parameters: n − estimators, max − depth, min − amples − split, and max − features. Other functions were performed according to the system default. In Python, the coarse value of the parameter was determined by Sklearn call RF Classifier firstly, and then refined by the GridSearchCV. Finally, the optimized parameter was imported into the classifier of eCognition, as shown in Table 4.

Then, all images were classified and trained by combining the VDVI and the sample information of each ground class, and then the training results were applied to all the images. Finally, the classified vegetation information was output in SHAP files, extracted and displayed in DOM, and analyzed quantitatively by calculating PA, UA, OA, and Kappa of the classification results.

Statistical RF classification algorithm based on vegetation, bare land, tail sand, and water showed the error matrix of samples of 84, 99, 60, and 8, respectively, as shown in Table 5. Vegetation sample data set with 1 sample was mistakenly identified as bare land, bare sample data set contained four samples were mistakenly identified as vegetation, and water body with a sample data set was mistakenly identified as tail ore. RF classification algorithm achieved 97.60% OA and 96.60% Kappa coefficient in this test. The higher the Kappa coefficient value was, the higher the classification accuracy was, so the classification effect was better.

Based on ArcGIS, four types of land information of DOM classification were read and screened according to attributes, and only vegetation information was retained. Then, the vegetation classification information was overlayed by RF algorithm to DOM, as shown in Figure 15. Large areas covered by vegetation in the research area could be identified based on the visual interpretation combined with the field survey. L_1_ and L_2_ did not misidentify the bare land embedded in the middle while recognizing vegetation information. However, bare soils of R_1_ in L_3_ and R_2_ and R_3_ in L_4_ were wrongly identified as vegetation information. After the DIM point cloud was mapped, bare soil information of wrongly identified vegetation was removed manually.

### 4.2. 2D Image Vegetation Information and 3D Point Cloud Mapping

The 2D images were classified based on plane shape. Due to sparse vegetation in the field, there was some bare soil information in the extracted vegetation. The DIM point clouds were obtained from UAV images through dense matching, the total number of point clouds was 384,410,757 points, and the DOM used for classification was constructed based on UAV images. Therefore, the direct linear transformation method was employed to extract vegetation information from the 2D images and to map the 3D point clouds, and the mapping error was counted according to Equation (2). There were 2786 pieces of planar information of all ground classes, and 1019 pieces of vegetation planar information were counted after the ground class information, such as bare ground, tailing sand, and water body, were excluded, which were mapped into the 3D point cloud. In addition, 998 pieces of planar vegetation information were counted into the 3D point cloud. Therefore, the mapping accuracy of this method was 97.93%. The number of point clouds included in the surface information was 144,095,774, and the mixed bare soil in the vegetation was eliminated in the form of points. Finally, the vegetation information was 138,618,519 points, accounting for 36.06% of the total number of points in the research area, as shown in Figure 16. The migration of radioactive elements in phosphate mining could affect the surface environment through phosphogypsum, waste slag, and waste liquid, etc. In the 3D point cloud construction of the image of the research area, spectral information of the vegetation in the lower part of the left area of tailings accumulation connected with the lawn dam was not obvious, which can be shown as wilting, and some of the affected area only had bare soil information. In the right part of tailings accumulation, the water content of waste raw materials was lower, the nuclide migration was less, and the vegetation information of lawn dam was more complete. For the direct solution methods of the mapping between 2D and 3D point clouds, domestic and foreign scholars have proposed the angular cone solution method [25], the collinear equation direct adjustment method [26], the Rodrigue matrix method [27], and the direct linear change method [10]. The next step is to study the differences in the mapping accuracy of tailing area of plateau phosphate mine by different methods.

## 5. Conclusions

Monitoring vegetation coverage in the tailing area of plateau phosphate mine was of great significance to evaluate the greening and restoration benefit. In this work, a DOM with 21,715 × 18,108 pixels and a resolution of 0.08 m was constructed from two aerial images. Based on the terrain factors, vegetation information was firstly segmented and extracted by combining visible VI and RF classification algorithm. Finally, 2D vegetation information was mapped to a 3D point cloud, and FVC was counted after elimination of the redundant data.

(1) UAV aerial photography was adopted to acquire the centimeter-level high-resolution images of vegetation in the plateau mining area, and 384,401,447 DIM point clouds were constructed, with a point cloud density of 214.36 points/m^2^, a low cost, and a high efficiency. (2) For 2D image segmentation, the local variances of segmentation objects were counted, and the preliminary scale interval was judged by the LV-ROC curve. Combined with the two terrain factors and the elevation, the optimal scale corresponding to the peak can be determined to be less than 500, and the optimal segmentation scale of this work was 440 based on the comprehensive slope factor. (3) Nine visible VIex were calculated by ArcGIS Pro, and the VDVI of the classified vegetation information of the tailing mining area was determined from the gray histogram of the index, ground error analysis, and qualitative and quantitative analysis of the bimodal index. (4) Based on eCognition and RF classification algorithm, the OA and Kappa coefficient of RF classification algorithm reached 97.60% and 96.60%, respectively, by selecting vegetation, bare land, tailing sand, and water samples in the research area. (5) Vegetation information was extracted from 2D image and mapped to the 3D point cloud, and redundant bare land information was eliminated in the form of surface to point. The statistical FVC was 36.06%. The smaller was the particle size of radioactive elements in tailings accumulation area, the more easily it was dissolved and absorbed by vegetation, and the absorption capacity of radionuclides of different vegetation was different. In the lawn dam area of the research area, vegetation such as sunflower and Sudan grass should be planted with better absorption of radioactive elements, so as not to expand the ecological damage caused by phosphate mining.

In this work, vegetation information was only extracted from 2D images and mapped to 3D point clouds, showing lower requirements for processing hardware but specific limitations. There were errors in mapping 2D image classification to 3D point clouds. Error evaluation of different mapping methods will be the focus of the next research. For the classification of 3D point clouds, there were some limitations such as long time consumption and high requirements on hardware equipment. For the DIM point clouds, how to utilize the spectral information in the DIM point clouds and how to quickly and conveniently extract the relevant information from the 3D perspective of DIM point clouds will also be related to the research work.

## Figures and Tables

**Figure 1 sensors-22-06388-f001:**
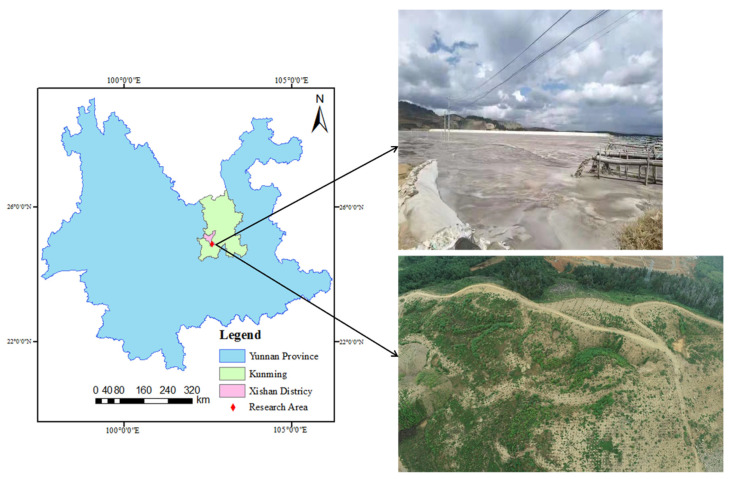
Position of research area.

**Figure 2 sensors-22-06388-f002:**
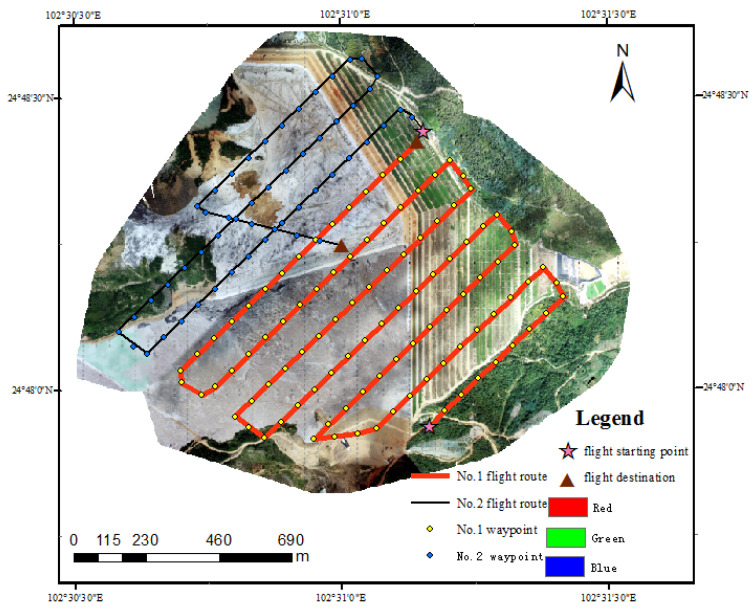
The specific route planning of the aerial survey.

**Figure 3 sensors-22-06388-f003:**
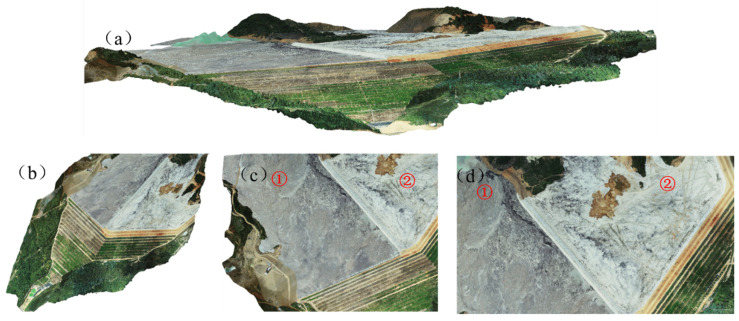
Construction of DIM point cloud. (**a**,**b**) the whole research area was completely constructed without point cloud holes. (**c**,**d**) disclosed that the tailings areas showed obvious contours. (1, 2 represent tailings areas have different water content).

**Figure 4 sensors-22-06388-f004:**
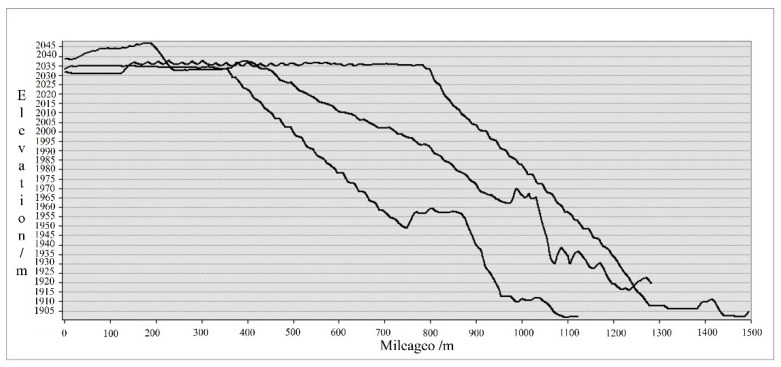
Sectional view.

**Figure 5 sensors-22-06388-f005:**
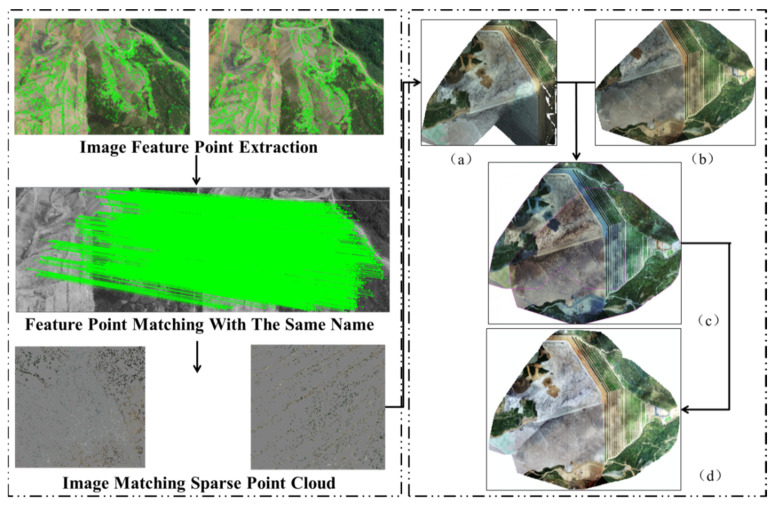
DOM construction process. (**a**,**b**) represents the result of the DOM construct. (**c**) represents the result of matching and fusion (**a**,**b**). (**d**) fusion chromatic difference was eliminated according to color space uniformity, and DOM was completely constructed.

**Figure 6 sensors-22-06388-f006:**
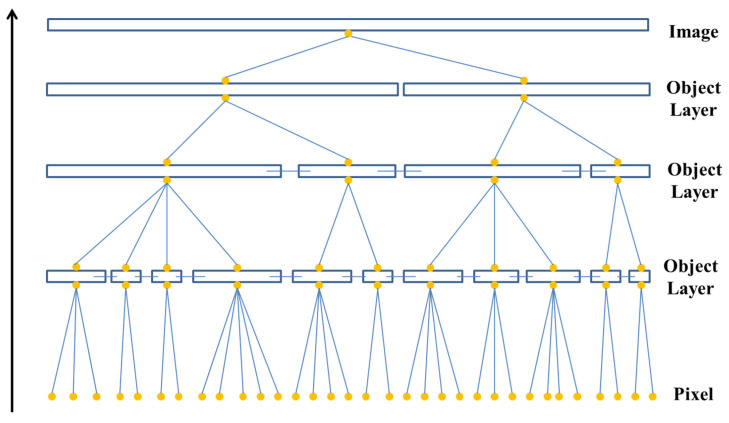
Multi-scale segmentation.

**Figure 7 sensors-22-06388-f007:**
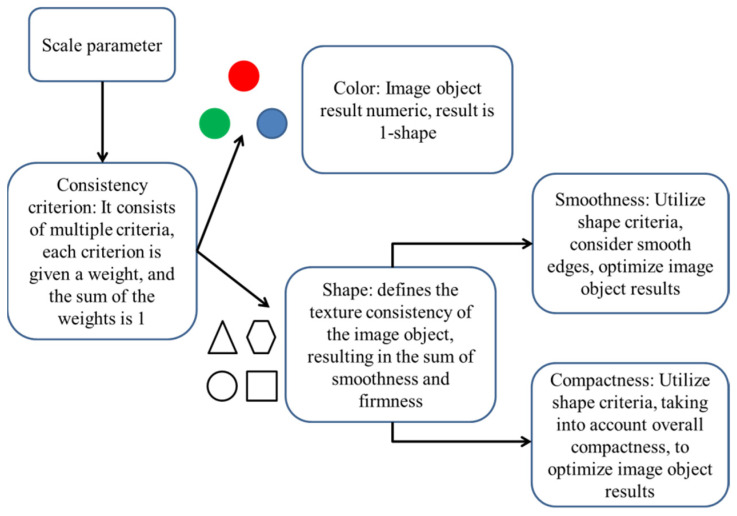
Main parameters of multi-scale segmentation.

**Figure 8 sensors-22-06388-f008:**
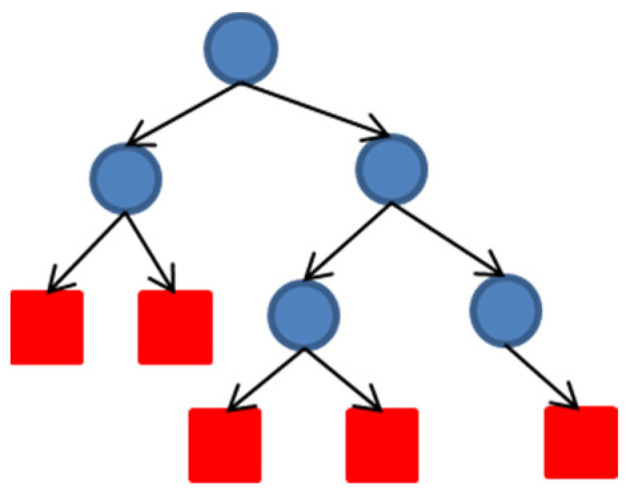
Illustration of decision tree.

**Figure 9 sensors-22-06388-f009:**
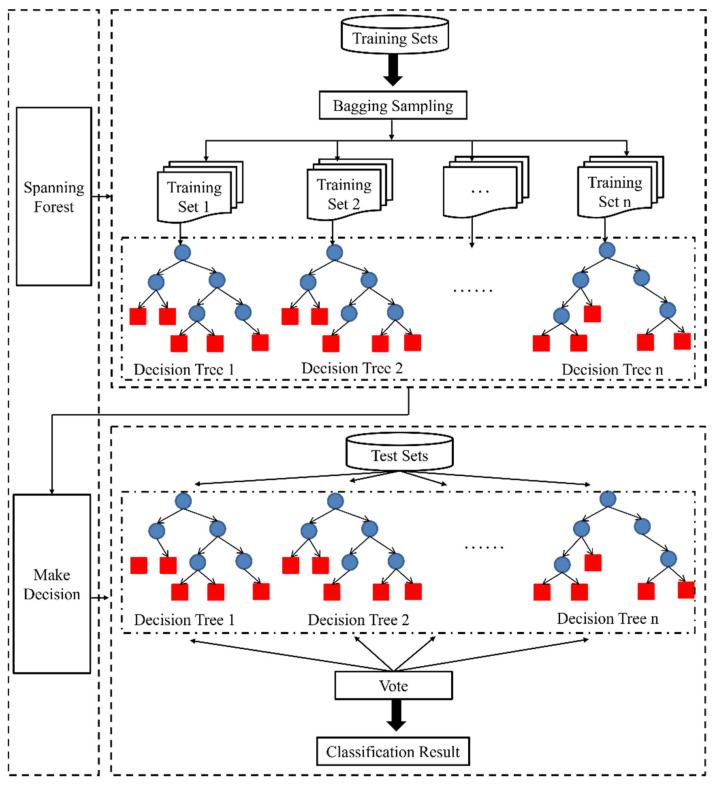
Random forest algorithm.

**Figure 10 sensors-22-06388-f010:**
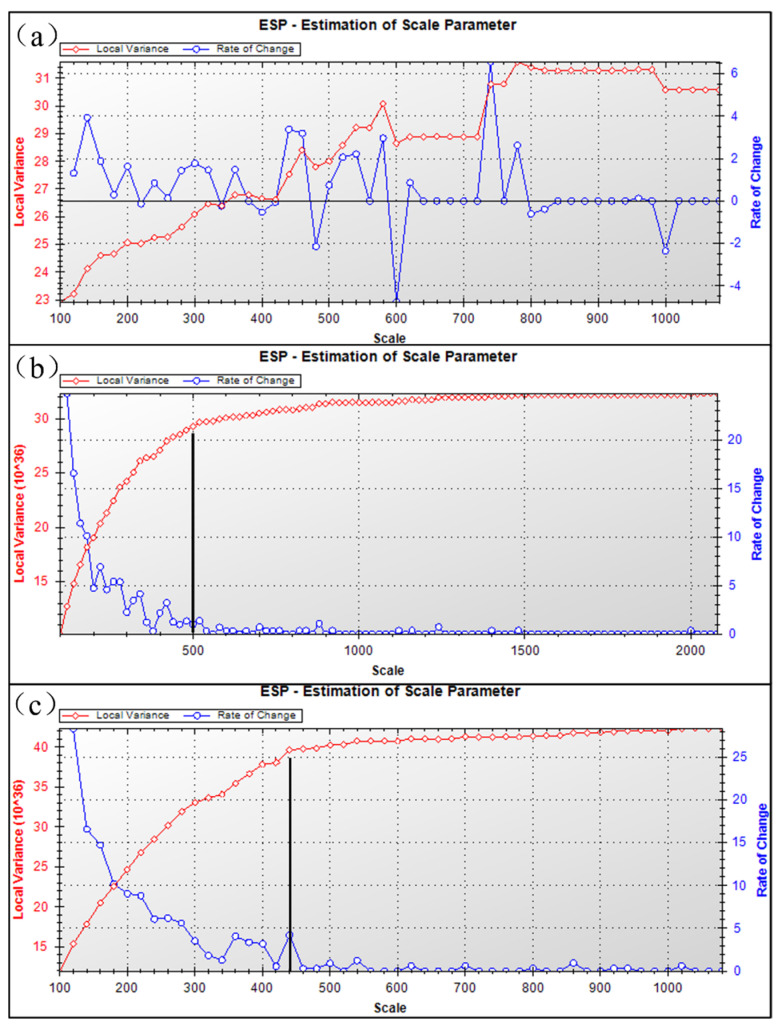
DOM segmentation scale. (**a**) represents the local variances of the segmentation object were counted to obtain the LV − ROC curve. (**b**) represents the LV − ROC curve obtained after the elevation was added. (**c**) represents the LV − ROC curve obtained after the slope was added.

**Figure 11 sensors-22-06388-f011:**
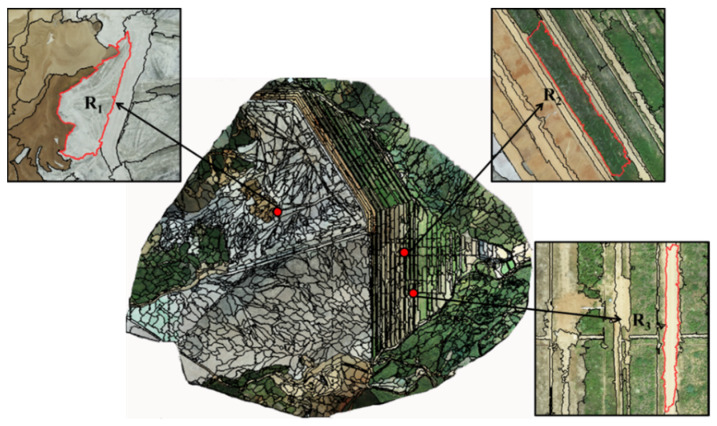
Segmentation results. R_1_ is the tailing sand and a small part of bare soil mixed area. R_2_ and R_3_ were the mixed distribution areas of vegetation and bare soil.

**Figure 12 sensors-22-06388-f012:**
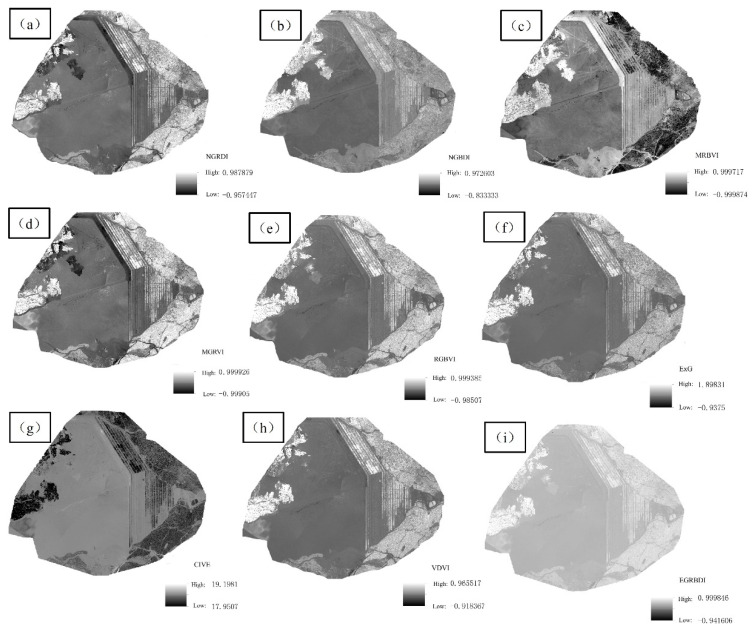
(**a**–**i**) Calculation results of each visible VI.

**Figure 13 sensors-22-06388-f013:**
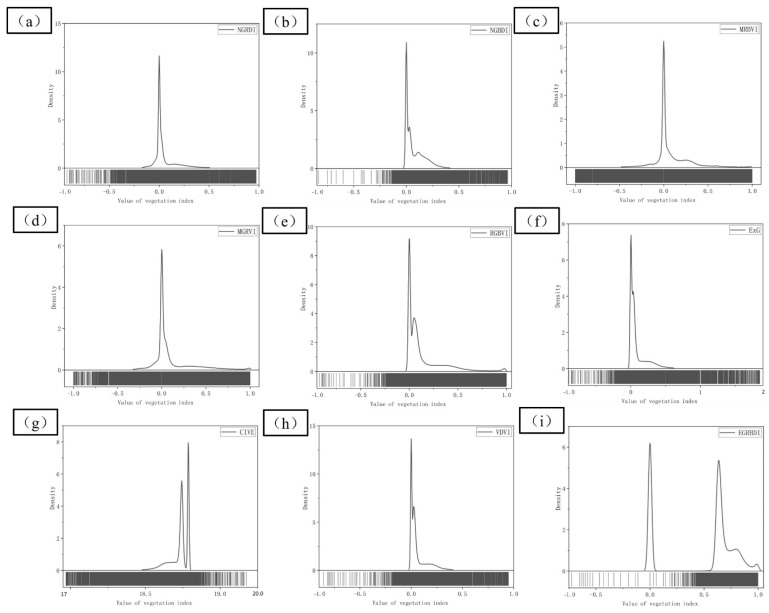
(**a**–**i**) Calculation results of statistical histogram each visible VI.

**Figure 14 sensors-22-06388-f014:**
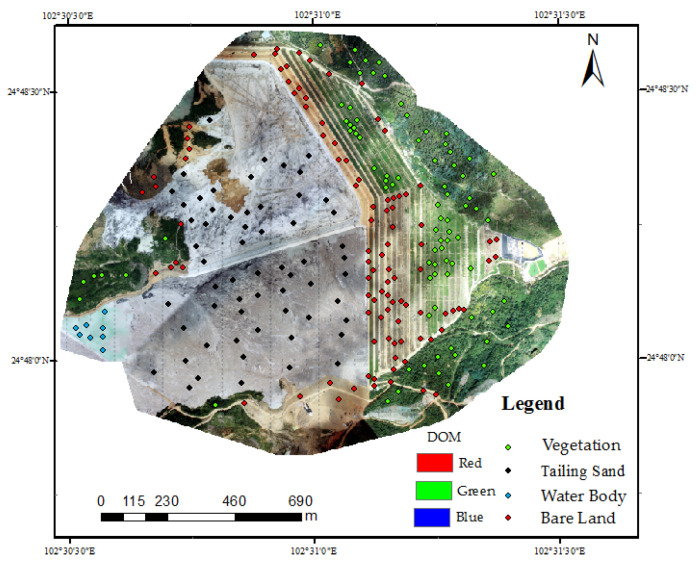
Sample point distribution.

**Figure 15 sensors-22-06388-f015:**
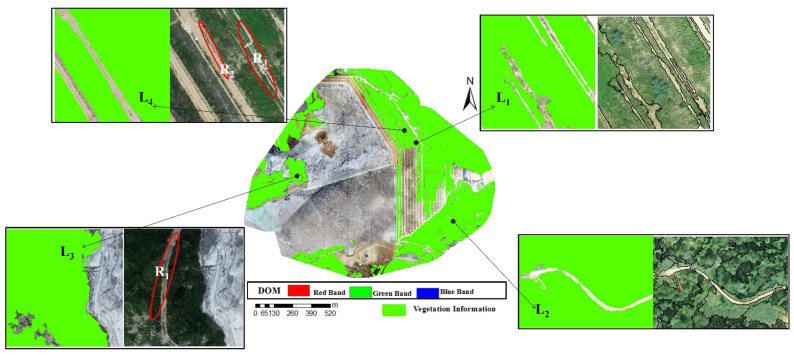
RF algorithm classification to extract vegetation information. L_1_ and L_2_ did not misidentify the bare land embedded in the middle while recognizing vegetation information. Bare soils of R_1_ in L_3_ and R_2_ and R_3_ in L_4_ were wrongly identified as vegetation information.

**Figure 16 sensors-22-06388-f016:**
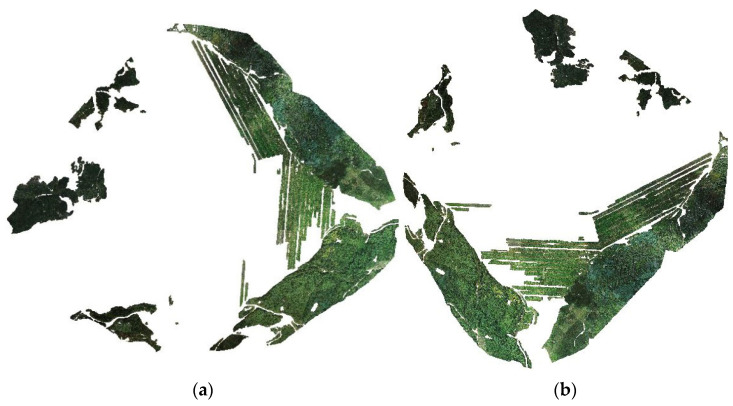
(**a**,**b**) represent DIM point cloud vegetation information.

**Table 1 sensors-22-06388-t001:** Comparison of DIM and LiDAR point clouds.

Comparative Indices	LiDAR Point Cloud	DIM Point Cloud
Data acquisition method	Lidar sweeps	UAV photography and image intensive matching
Influence factors	Scanning mode and actual occlusion	Image quality and matching accuracy
Point cloud data integrity	Generally, there is an object block easy to appear empty	Preferably
Color information	Without	Possess
Density	Low	High
Standard Deviation	Little	Big
Echo frequency and intensity	Possess	Without
Scanline information	Possess	Without

**Table 2 sensors-22-06388-t002:** Combined VIex in visible bands.

Name	VI	Equation	Range Theory
Normalized green-red difference index	NGRDI [18]	$\frac{G-R}{G+R}$	[−1, 1]
Normalized green-blue difference index	NGBDI [19]	$\frac{G-B}{G+B}$	[−1, 1]
Modified red and blue vegetation index	MRBVI [15]	$\frac{R^{2}-B^{2}}{R^{2}+B^{2}}$	[−1, 1]
Modified green red index	MGRVI [20]	$\frac{G^{2}-R^{2}}{G^{2}+R^{2}}$	[−1, 1]
Red, green, and blue vegetation index	RGBVI [21]	$\frac{G^{2}-B\times R}{G^{2}+B\times R}$	[−1, 1]
Extreme green index	ExG [22]	2g−r−b	[−1, 2]
Color index of vegetation extraction	CIVE [23]	0.44r−0.88g+0.385b+18.78745	[17, 20]
Visible band difference vegetation index	VDVI [17]	$\frac{2G-R-B}{2G+R+B}$	[−1, 1]
Excess green–red–blue difference index	EGRBDI [21]	$\frac{{\left(2G\right)}^{2}-B\times R}{{\left(2G\right)}^{2}+B\times R}$	[−1, 1]

Note: *R*, *G*, and *B* referred to red, green, and blue visible bands, respectively; *r*, *g*, and *b* were standardized results of *R*, *G* and *B*, which were expressed as: $r=\frac{R}{R+G+B}$
$g=\frac{G}{R+G+B}$, and $b=\frac{B}{R+G+B}$, respectively.

**Table 3 sensors-22-06388-t003:** Error statistics of four land types of VI.

Name	Error	Vegetation	Bare Land	Tailing Sand	Water Body	Range Value
NGRDI	Mean Std	0.01 0.085	−0.05 0.027	0.01 0.013	0.12 0.08	[−1, 1]
NGBDI	Mean Std	0.12 0.084	0.15 0.034	0.03 0.014	0.05 0.08	[−1, 1]
MRBVI	Mean Std	0.23 0.145	0.39 0.105	0.04 0.053	−0.14 0.144	[−1, 1]
MGRVI	Mean Std	0.01 0.148	−0.10 0.053	0.03 0.057	0.23 0.147	[−1, 1]
RGBVI	Mean Std	0.13 0.128	0.10 0.021	0.05 0.007	0.16 0.127	[−1, 1]
ExG	Mean Std	0.08 0.122	0.05 0.014	0.03 0.005	0.11 0.121	[−1, 1]
CIVE	Mean Std	18.7 0.052	18.7 0.006	18.8 0.002	18.7 0.05	[18, 19]
VDVI	Mean Std	0.04 0.075	0.04 0.010	0.02 0.003	0.08 0.07	[−1, 1]
EGRBDI	Mean Std	0.68 0.063	0.66 0.012	0.63 0.004	0.69 0.06	[0, 1]

**Table 4 sensors-22-06388-t004:** Main parameters of RF algorithm.

Name	n − Estimators	max − Depth	max − Features	min − Samples − Split
Coarse values	71	15	0.1	2
Optimized values	75	16	0.1	3

**Table 5 sensors-22-06388-t005:** Classification accuracy statistics of RF algorithm.

Error Matrix Based on Sample	Vegetation	Bare Land	Tailing Sand	Water Body	Total
Vegetation	83	1	0	0	84
Bare land	4	95	0	0	99
Tailing sand	0	0	60	0	60
Water body	0	0	1	7	8
Total	87	96	61	7	
Accuracy					
PA	0.954	0.989	0.983	1	
UA	0.988	0.959	1	0.875	
OA	0.976				
Kappa	0.965				

## Data Availability

Not applicable.

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
