# Peer review of "Test and Analysis of Vegetation Coverage in Open-Pit Phosphate Mining Area around Dianchi Lake Using UAV–VDVI"

_sensors, 2022, doi:10.3390/s22176388_

Round 1

Reviewer 1 Report

(104~106) Figure 1, yunnan Map is not in correct ratio.

(160-162) Figure 4, What is the meaning of three curves in the figure?

(259-260) Figure.9 Random forest classification algorithm is not clear in evolution

(350~351) Figure 10 is not clear. 

All figures are very poor both in physics and analysis. 

Reviewer 2 Report

Please correct the way you insert the citations of the references. One image resolution should be improved. The work seems well done

Reviewer 3 Report

The manuscript describes an approach to test and analyze the vegetation coverage in open-pit phosphate mining area around Dianchi Lake. However, the current manuscript is not publishable. 

Main comments: 

1) There are too many grammar issues throughout the manuscript. Please resubmit the manuscript after extensive lanuage edits. 

2) The manuscript looks more like an engineering report instead of a scientific paper. For example, the authors mostly focus on describing the proposed system using plain language, say Section 3.1. If the reviewer has understood it correctly, the segmntation and classification should be one of the core components in this study. The authors need to provide technical details of the system in mathematical form instead of using plain language. 

3) The contribution of this work seems weak for a publication in this journal. The system is built upon the available bricks. Where is the novel computational contribution? Only mentioning the available tools such as the image segmentation and random forest for classification (both are sort of common knowledge) is not sufficient to result in a publication. 

4) As for the Section 4.2, the reviewer wonders where is the quantified results for the mapping? How do the authors evaluate the mapping performance? Only showing the mapped results visually is not enough. Please find a suitable metric and report the quantified performance. 

Other comments:

1) The title is worded strangely. Please consider changing the title. 

The suggested title should be like: Test and Analysis of Vegetation Coverage in Open-Pit Phosphate Mining Area around Dianchi Lake Using UAV-VDVI

2) The abstract needs rewrite. The current abstract provides too many details of the proposed system and the experimental results. 

Generally, the Abstract is used as a high-level introduction of the work, and the authors should summarize the content. Please revise. 

3) For easy readability, the reviewer recommends the authors highlighting the contributions of this study as bullet point at the end of the introduction part. 

Round 2

Reviewer 3 Report

The reviewer thanks the authors to make such changes. 

The only comments: 

1) The writeup of the Section 3.1 is cryptic. The technical details of segmentation and classification are missing. For example, what is the input and output? What is the training process? What is the objective? What is the loss function? How does the optimization work to output the required parameters? Etc. 

Without mentioning these details, how do the authors expect the others to reproduce the work? Please revise. 

2) As suggested in point 7, the authors need to highlight the contributions of this work (response of point 3) at the end of introduction to make it visible to the readers.  

3) The font from L94-L97 (and maybe other lines, please check) seems different to the others, please make them consistent. 
